# Evolution of Systemic Therapy in Medulloblastoma Including Irradiation-Sparing Approaches

**DOI:** 10.3390/diagnostics13243680

**Published:** 2023-12-16

**Authors:** Naureen Mushtaq, Rahat Ul Ain, Syed Ahmer Hamid, Eric Bouffet

**Affiliations:** 1Division of Pediatric Oncology, Department of Oncology, Aga Khan University, Karachi 74800, Pakistan; naureen.mushtaq@aku.edu; 2Department of Pediatric Hematology/Oncology & Bone Marrow Transplant, University of Child Health Sciences, Children’s Hospital, Lahore 54600, Pakistan; dr.rkashif@yahoo.com; 3Department of Pediatric Hematology and Oncology, Indus Hospital & Health Network, Karachi 74800, Pakistan; sahmerhamid@gmail.com; 4Global Neuro-Oncology Program, Department of Global Pediatric Medicine, St. Jude Children’s Research Hospital, St. Jude Global, Memphis, TN 38105, USA

**Keywords:** medulloblastoma, subgrouping, chemotherapy, high-dose chemotherapy

## Abstract

The management of medulloblastoma in children has dramatically changed over the past four decades, with the development of chemotherapy protocols aiming at improving survival and reducing long-term toxicities of high-dose craniospinal radiotherapy. While the staging and treatment of medulloblastoma were until recently based on the modified Chang’s system, recent advances in the molecular biology of medulloblastoma have revolutionized approaches in the management of this increasingly complex disease. The evolution of systemic therapies is described in this review.

## 1. Introduction

Described for the first time nearly 100 years ago, medulloblastoma is the most common malignant brain tumor in the pediatric population. In their seminal report in 1925, Percival Bailey and Harvey Cushing described 29 medulloblastoma patients with only 4 radical resections (14%) and 2 survivors (7%) at the time of the publication [1]. The management of medulloblastoma has largely evolved over the past century and has been transformed with the introduction of craniospinal irradiation in the 1950s and chemotherapy in the 1970s. Currently, 5-year survival rates for patients with average-risk medulloblastoma in high-income countries are above 80%, while they range between 60 and 70% for patients with high-risk disease [2].

Over the past decade, advancements in molecular classification have provided new insights into the landscape of this complex disease. The discovery of four different subgroups (Sonic Hedgehog, WNT, subgroup 3, and subgroup 4) has been a critical milestone, opening the prospect of new clinical trials based on the molecular profile [3,4]. However, the implementation of these protocols has been relatively slow, and most patients are still treated according to the modified Chang classification based on the extent of postoperative residual tumor and subarachnoid dissemination [5].

Recent protocols in North America and in Europe have contributed to the further refinement of medulloblastoma into molecular subtypes, offering the opportunity to identify specific groups of patients who may benefit from treatment de-escalation, sparing them adverse effects, but also to select patients who may be candidates for more intensive treatments. In this context, protocols for medulloblastoma patients are becoming increasingly complex and will require national/international collaboration to achieve their objectives. This review summarizes the current knowledge of systemic therapies for medulloblastoma.

## 2. The Subgrouping of Medulloblastoma

Medulloblastoma is composed of four distinct molecular and clinical variants: WNT, Sonic Hedgehog (SHH), group 3, and group 4 (often termed non-WNT, non-SHH). The WHO 2016 classification for CNS tumors additionally divided SHH medulloblastomas according to their TP53 status due to their different clinicopathological characteristics [6]. Methylation and “omics” studies have further identified new subgroups and currently, there are two different WNT subgroups, four different SHH subgroups, and eight subgroups of non-WNT/non-SHH medulloblastoma [7,8]. While gene expression and methylation profiling are the gold standard for defining the molecular groups of medulloblastoma, they are not universally accessible. In the absence of these techniques, immunohistochemistry assays can be used to identify specific subgroups. GAB1 characterizes only SHH tumors and nuclear immunoreactivity for β-catenin only WNT tumors. There is currently no reliable immunohistochemistry assay for group 3 and 4 tumors [9]. This new classification of medulloblastoma has recently been integrated in clinical trials, and the management of this disease is becoming increasingly complex and technical. Importantly, there is a before and an after. Most clinical trials conducted before the period 2000–2010 did not include subgrouping, and the comparison of these trials with most recent ones is therefore challenging.

## 3. Principles of the Management of Pediatric Medulloblastoma

The first step in the management of medulloblastoma is surgery. The importance of the resection has been established [10], and the recommendation is to achieve a complete “safe” resection. The value of this statement has been re-assessed in the context of the subgrouping of medulloblastoma, suggesting that there is no definitive benefit to gross total resection (GTR) compared with near-total resection, when the likelihood of neurological morbidity is high in achieving GTR [11]. Postoperative treatment in the pediatric population differs according to the age of the patient. Due to the devastating consequences of craniospinal irradiation in infants and young children, there is a general consensus to use a radiation-sparing approach in this age group. Postoperative treatment is therefore based on intensive/high-dose chemotherapy associated in some protocols with intraventricular injections of chemotherapy via an Ommaya reservoir [12]. In older children, postoperative treatment consists of a combination of craniospinal irradiation and chemotherapy. The cut-off between young and older children has traditionally been at the age of 3 [13]. However, recently, several protocols and cooperative groups have considered different cut-offs, either at the age of 4 [14], 5 [15], and even 6 [16].

## 4. Evolution of Systemic Therapies in Medulloblastoma

The first reports on chemotherapy can be found in the early 1960s, when vincristine, methotrexate (intrathecal and systemic), and nitrosourea were introduced in the management of recurrent and newly diagnosed brain tumors, including medulloblastoma [17]. In the absence of modern imaging, the benefit of these treatments was essentially assessed clinically [18]. The development of large clinical trials by the SIOP (International Society of Paediatric Oncology) and the CCG (Children’s Cancer Group) was the first attempt to systematically evaluate the impact of chemotherapy in medulloblastoma [19,20]. Both SIOP1 and CCG942 trials had a similar design, comparing craniospinal irradiation (CSI) with a boost to the posterior fossa and metastatic deposits with or without the addition of chemotherapy. The SIOP1 protocol used a combination of CCNU and vincristine, whereas the CCG942 trial used prednisone in addition to CCNU and vincristine. These trials failed to demonstrate a significant benefit of nitrosourea-based chemotherapy. However, subgroup analyses identified a benefit of the addition of chemotherapy in patients with bulky or advanced disease in the CCG trial [19]. These trials also contributed to identify prognostic factors that were eventually used for stratification in subsequent clinical trials. The introduction of platinum compounds in the management of medulloblastoma has been associated with a significant increase in survival rates, as illustrated in the SEER Statistical Report [21]. However, this finding may also be the joint result of improved surgical management, an improvement in radiotherapy techniques, as well as the incorporation of intensive chemotherapy protocols into clinical practice. Progressively, the management of medulloblastoma has witnessed the emergence of two different approaches, one for older children, with the use of craniospinal irradiation associated with multiagent chemotherapy, and one for younger children, who are treated with a chemotherapy-only and radiation-sparing strategy. Over the past decades, several clinical trials have been conducted in young and older children using different regimens. However, the discovery of four subgroups of medulloblastomas, with a further refinement in the subgrouping using molecular and cytogenetic data, has made the interpretation of the results of old protocols difficult and obsolete. We are currently witnessing an extensive re-assessment of classical risk criteria in the context of this subgrouping [4,22]. As a result, the concept of high-risk factors such as residual disease, anaplasia, or even metastatic disease has recently been challenged [4,11,23].

As far as chemotherapy agents are concerned, protocols for older children in the upfront setting have shown little change and include a limited number of agents, mostly cisplatin, cyclophosphamide, vincristine, and lomustine in North America [24,25], whereas European protocols also use etoposide, carboplatin, ifosfamide, and high-dose methotrexate [26,27,28]. In younger children, in addition to these drugs, thiotepa is often part of high-dose chemotherapy regimens [29]. Protocols for young children and in some cooperative groups for metastatic patients use repeated injections of intraventricular chemotherapy [26]. 

## 5. Current Considerations for Older Medulloblastoma Patients

Average risk medulloblastoma patients (Table 1)

**Table 1 diagnostics-13-03680-t001:** Average risk trials.

Average Risk MB	Population	Number	Dose of Craniospinal/ Fractionation	Dose Posterior Fossa or Tumor Bed	Chemotherapy Regimen	5-Year PFS and/or OS
Protocol (Reference)						
HIT91 [30,31]	M0	45	35.2 Gy/1.6 Gy	Boost 20 PF Gy/2.0 Gy	RT followed by maintenance chemotherapy with lomustine, cisplatin, and vincristine (eight cycles).	5-year PFS 83%
	M0	69	35.2 Gy/1.6 Gy	Boost PF 20 Gy/2.0 Gy	“Sandwich chemotherapy” with ifosfamide, etoposide, high-dose MTX, cisplatin, and cytarabine (one or two cycles according to response) before RT followed by maintenance chemotherapy after RT with lomustine, cisplatin, and vincristine if incomplete response.	5-year PFS 53%
PNET3 [32]	M0/1	90	35 Gy/1.67 Gy	Boost PF up to 55 Gy/1.67 Gy	Pre-radiotherapy chemotherapy with alternating courses of vincristine; carboplatin, etoposide, and vincris-tine, and etoposide and cyclophos-phamide (a total of four cycles).	5-year PFS 74.2%
	M0/1	89	35 Gy/1.67 Gy	Boost PF up to 55 Gy/1.67 Gy	No chemotherapy.	5-year PFS 59.8%
COG9961 [24]	M0	193	23.4 Gy/1.8 Gy	Boost PF up to 55.8 Gy/1.8 Gy	Six doses of vincristine (1.5 mg/m^2^) during radiotherapy; eight cycles of cisplatin (70 mg/m^2^) lomustine (75 mg/m^2^) and vincristine (1.5 mg/m^2^ day 1, 8 and 15) post radiotherapy.	5-year PFS 80% 5-year OS 85%
	M0	186	39.6 Gy/1.8 Gy	Boost PF up to 55.8 Gy/1.8 Gy	Six doses of vincristine (1.5 mg/m^2^) during radiotherapy; eight cycles of cisplatin (70 mg/m^2^) cyclophosphamide (1000 mg/m2 Day 1 and 2) and vincristine (1.5 mg/m^2^ day 1, 8 and 15) post radiotherapy.	5-year PFS 82% 5-year OS 87%
SJMB96 [33]	M0	86	23.4 Gy/1.8 Gy	Boost PF up to 36 Gy/1.8 Gy and TB up to 55.8 Gy/1.8 Gy	Four cycles of post radiotherapy chemotherapy with cyclophosphamide (4 g/m^2^), cisplatin (75 mg/m^2^), and vincristine (1.0 mg/m^2^) with autologous stem cell rescue.	5-year PFS 83%
PNET4 [34]	M0	169	23.4 Gy/1.8 Gy	Boost TB up to 55.8 Gy/1.8 Gy	Eight doses of vincristine (1.5 mg/m^2^) during radiotherapy; eight cycles of cisplatin (70 mg/m^2^) lomustine (75 mg/m^2^) and vincristine (1.5 mg/m^2^ on day 1, 8, and 15) post radiotherapy.	5-year PFS 78%
	M0	169	36 Gy/1.0 Gy (BID)	Boost posterior fossa up to 60 Gy/1.0 Gy (BID) and up to 68 Gy/1.0 Gy (BID) tumor bed		5-year PFS 81%
ACNS0331 [25]	M0	110	23.4 Gy/1.8 Gy	Boost PF or TB	Six doses of vincristine (1.5 mg/m^2^) during radiotherapy; six cycles of (A) cisplatin (70 mg/m^2^), lomustine (75 mg/m^2^), and vincristine (1.5 mg/m^2^ on day 1, 8, and 15), and three cycles of (B) vincristine (1.5 mg/m^2^ on day 1 and 8), and cyclophosphamide (1000 mg/m^2^ on day 1 and 2) post radiotherapy, with a cycle schedule AABAABAAB.	5-year PFS 82.9% (23.4 Gy) versus 71.4% (18 Gy)
	M0	116	18 Gy/1.8 Gy	Boost PF or TB		
	M0	227	18 or 23.4 Gy/1.8 Gy	TB boost up to 55.8 Gy/1.8 Gy		5-year PFS 82.5% (TB boost) versus 80.5% (PF boost)
	Mo	237	18 or 23.4 Gy/1.8 Gy	PF boost up to 55.8 Gy/1.8 Gy		
SJMB03 [23]	M0	227	23.4 Gy/1.8 Gy	Boost to tumor bed up to 55.8 Gy/1.8 Gy	Four cycles of post-radiotherapy chemotherapy with cyclophosphamide (4 g/m^2^), cisplatin (75 mg/m^2^), and vincristine (1.0 mg/m^2^) with autologous stem cell rescue.	5-year PFS 82.3% 5-year OS 88%

PFS: progression free survival; OS: overall survival. Gy: Gray; PF posterior fossa; TB: tumor bed; MTX: methotrexate.

Classically, the term average risk medulloblastoma defined a situation where the patient did not show evidence of metastatic disease and underwent a complete or subtotal resection, with a size of residual ≤1.5 cm^2^. In some cooperative groups, an additional requirement concerns the absence of anaplasia. However, recent evidence shows that the prognostic value of anaplasia is group-specific, and the presence of anaplasia does not affect survival in group 4 patients [23]. 

For this group of patients, the consensus is to proceed to upfront craniospinal radiotherapy, if possible, within 4 to 5 weeks following surgical resection. Most protocols include a weekly administration of vincristine during the 6 weeks of radiotherapy, although this practice has been abandoned in the SJMB protocols without clear evidence of negative impact on survival [23,33]. Post-radiation chemotherapy consists of several cycles of chemotherapy. Protocol 9961 was a randomized trial comparing eight cycles of cisplatin, CCNU, and vincristine versus eight cycles of cisplatin, cyclophosphamide, and vincristine [24]. This trial has been pivotal in demonstrating the feasibility and safety of reduced-dose craniospinal radiotherapy in a selected group of patients with average-risk features. While before 2000, most protocols in Europe and North America were considering a craniospinal dose of 36 Gy for all patients [10,32], regardless of their staging characteristics, protocol 9961 has been the trigger for a new generation of protocols for average-risk patients treated with doses of craniospinal radiotherapy of 18–23.4 Gy [25,34,35]. In protocol 9961, there was no significant difference in survival between the two chemotherapy regimens. However, ototoxicity was identified as a major concern [36], and the subsequent protocol, ACNS0331, reduced the number of courses of cisplatin-CCNU and vincristine to six cycles (A), with the addition of three cycles of vincristine and cyclophosphamide (B) in a AABAABAAB sequence [25]. The ongoing protocol of the Children’s Oncology Group, ACNS2031 (NCT05382338) (https://clinicaltrials.gov/study/NCT05382338, accessed on 19 September 2023), is using the same schedule, with the introduction of sodium thiosulfate, with the aim to reduce the risk of ototoxicity. 

The SJMB group has run three consecutive trials over the last 25 years. SJMB96 and SJMB03 had a similar design for average-risk patients and consisted of four consecutive courses of high-dose chemotherapy with stem cell rescue one month apart, administered after craniospinal irradiation. The cumulative dose of cisplatin was 300 mg/m^2^, while the cumulative dose of cyclophosphamide was 16 g/m^2^. This approach showed a 5-year progression-free survival rate of 83.2% in average-risk patients, while the 5-year progression-free survival rate of patients treated with standard dose CSI in ACNS0331 was 82.9% [23,37]. However, SJMB was reporting more concerning side effects, including a high risk of ovarian failure in a high proportion of female patients [38]. SJMB12 (NCT01878617) (https://clinicaltrials.gov/study/NCT01878617, accessed on 19 September 2023) has a complex design, based on tumor subgrouping and risk stratification. In this protocol, patients with WNT and SHH medulloblastoma receive four cycles of cyclophosphamide, cisplatin, and vincristine. Post-pubertal patients with SHH medulloblastoma receive a Vismodegib maintenance. Average-risk patients with non SHH-non WNT medulloblastoma receive four to seven cycles of chemotherapy. Patients with predefined high-risk features (MYC or MYCN amplification) receive three cycles of gemcitabine and pemetrexed, in addition to four cycles of vincristine-cyclophosphamide-cisplatin. The use of the gemcitabine-pemetrexed combination is based on preclinical data [39]; however, no phase II trial has demonstrated its activity in newly diagnosed or recurrent medulloblastoma. The final results of SJMB12 are pending. 

Due to the excellent prognosis of patients with WNT medulloblastoma, two pilot studies have attempted to avoid the use of craniospinal radiotherapy. One protocol used adjuvant chemotherapy as per the protocol ACNS0331, and recruited six patients [40]. The first two enrolled patients experienced early local and leptomeningeal relapse and were salvaged with craniospinal irradiation. In view of these early failures, two patients received radiotherapy immediately after completing chemotherapy. One patient was removed from the study to receive intensification with high-dose chemotherapy. A sixth patient experienced early relapse and succumbed to disease progression despite craniospinal irradiation. Another clinical trial, conducted at Tata Memorial Hospital (India), evaluated the efficacy of focal radiotherapy to the tumor bed followed by six cycles of cisplatin, cyclophosphamide, and vincristine [41]. The trial was closed after 2 years, after three of the seven enrolled patients experienced disseminated relapse. Both studies suggest that conventional chemotherapy only is not an alternative to craniospinal irradiation, even in very good-risk WNT patients. Ongoing studies in Europe and North America are exploring the possibility to reduce the dose of craniospinal irradiation to 18 or even 15 Gy in average-risk WNT medulloblastoma patients. 

Another interesting observation concerning WNT patients concerns the type of chemotherapy administered. In a review of 93 molecularly confirmed WNT medulloblastomas, Nobre et al. identified 15 relapses. Maintenance chemotherapy with high cumulative doses of cyclophosphamide was a significant predictor of improved survival, suggesting that in this subgroup, the type of chemotherapy administered deserves careful consideration [42].

The SHH subgroup accounts for approximately 30% of all medulloblastomas. In SHH medulloblastoma, there is a constitutive activation of hedgehog signaling, often due to inactivating mutations of PTCH1. The inhibition of the hedgehog pathway by smoothened homologue inhibitors such as cyclopamine and HhAntag results in the regression of medulloblastoma tumors in PTCH1 mutant mice. This led to the development of SHH inhibitors, in particular vismodegib and sonidegib [43,44]. When the antitumor activity of these inhibitors in SHH medulloblastoma patients was first reported, there was a great hope that the introduction of these agents would be the first step of a progressive transition toward the use of targeted therapies. However, the overall response rate in clinical trials has been relatively modest [44,45] and the bone toxicity of SHH inhibitors, particularly the risk of irreversible growth plate fusion, precludes their use in prepubertal children [46]. Attempts are ongoing to develop alternative routes of administration to minimize or avoid these side effects [47]. 

High-risk medulloblastoma patients (Table 2)

**Table 2 diagnostics-13-03680-t002:** High-risk trials.

	Risk Category	Number	Dose of Craniospinal/ Fractionation	Dose Posterior Fossa	Chemotherapy Regimen	5 Year-PFS
Protocol						
CCG921 [10]	M1	31	36 Gy/1.8 Gy	Boost 18 Gy/1.8 Gy	Randomization: two cycles of eight drugs in one day before RT and eight cycles after versus eight cycles or weekly vincristine during XRT and eight cycles of vincristine, lomustine, and prednisone after XRT.	57.0%
	M2	12	36 Gy/1.8 Gy	Boost 18 Gy/1.8 Gy		40.0%
	M3	37	36 Gy/1.8 Gy	Boost 18 Gy/1.8 Gy		
POG9031 [48]	M1	29	35.5 Gy/1.6 Gy	Total dose 54.4 Gy	Randomization: three cycles of cisplatin-etoposide before or after RT; consolidation: seven cycles of cyclophosphamide-vincristine.	64.9%
	M2	36	40 Gy/1.6 Gy	Total dose 54.4 Gy		69.2%
	M3	34	40 Gy/1.6 Gy	Total dose 54.4 Gy		61.6%
HIT91 [30]	M1		35.2 Gy/1.6 Gy	Boost 20 Gy/2.0 Gy	Randomization “sandwich chemotherapy” with ifosfamide, etoposide, high-dose MTX, cisplatin, cytarabine (one or two cycles according to response) before RT followed by maintenance chemotherapy after RT with lomustine, cisplatin, and vincristine if incomplete response versus RT followed by maintenance chemotherapy with lomustine, cisplatin, and vincristine (eight cycles).	
	M2		35.2 Gy/1.6 Gy	Boost 20 Gy/2.0 Gy		
	M3		35.2 Gy/1.6 Gy	Boost 20 Gy/2.0 Gy		62.5%
POG9631 [49]	M1	11	39.6 Gy/1.8 Gy	Boost 16.2/1.8 Gy	Daily etoposide (day 1–21 and 29–49) during radiation. Maintenance with three courses of cisplatin-etoposide, followed by eight courses of cyclophosphamide and vincristine.	All patients 70.2%
	M2	6	39.6 Gy/1.8 Gy	Boost 16.2/1.8 Gy		M0 91.7%
	M3	18	39.6 Gy/1.8 Gy	Boost 16.2/1.8 Gy		M1 62.7% No data on M2-3
CCG99701 [50]	M1	18	36 Gy/1.8 Gy	Boost 19.8 Gy/1.8 Gy	Vincristine and phase I dose escalation of carboplatin during radiation Six cycles of post-radiotherapy chemotherapy with cyclophosphamide and vincristine (Regimen A) or cyclophosphamide, vincristine, and cisplatin (Regimen B).	77.0%
	M2	10	36 Gy/1.8 Gy	Boost 19.8 Gy/1.8 Gy		50.0%
	M3	49	36 Gy/1.8 Gy	Boost 19.8 Gy/1.8 Gy		67.0%
SJMB96 [33]	M1	9	36 Gy/1.8 Gy	Boost 19.8 Gy/1.8 Gy	Four courses of post-radiotherapy chemotherapy with high-dose cyclophosphamide, vincristine, and cisplatin with autologous stem cell transplant.	66.0%
	M2	6	39.6 Gy/1.8 Gy	Boost 16.2/1.8 Gy		
	M3	27	39.6 Gy/1.8 Gy	Boost 16.2/1.8 Gy		
SFOP [51]	M1	25	30.6 or 36 Gy/18 Gy	Boost up to 54 Gy/1.8 Gy	Two courses of eight drugs in one day and two courses of etoposide-carboplatin before radiotherapy; two cycles of eight drugs in one day (course 1 and 3) and etoposide-carboplatin (course 2 and 4) after radiotherapy.	56.0%
	M2/3	63	30.6 or 36 Gy/1.8 Gy	Boost up to 54 Gy/1.8 Gy		44.2%
HIT2000 [52]	M1	35	40 Gy/1 Gy BID	Boost 28 Gy/1 Gy BID	Two cycles of cyclophosphamide-vincristine, 2 × HDMTX-vincristine, carboplatin-etoposide, and intraventricular MTX before radiotherapy; four courses of maintenance chemotherapy with lomustine, cisplatin, and vincristine post radiotherapy.	61.0%
	M2/3	84	40 Gy/1 Gy BID	Boost 28 Gy/1 Gy BID		60.0%
HART Milan [53]	M1	9	31.2 Gy (if CR and age <10 years old) or 39 Gy (other cases)/1.3 Gy BID	Boost up to 59.7 ((less than 10 years old) to 60 Gy/1.5 Gy BID	Four courses of high-dose MTX (course 1), high-dose etoposide (course 2), high-dose cyclophosphamide (course 3), and carboplatin (course 4) before XRT; if complete remission before RT, maintenance with six courses of lomustine-vincristine; if no complete response, intensification with two courses of high-dose thiotepa.	78.4%
	M2	6				70.0%
	M3	17				
HART CCLG [54]	M1	9	39.68 Gy/1.24 Gy BID	22.32 Gy/1.24 Gy BID	Weekly vincristine during RT; maintenance: eight cycles of lomustine, cisplatin, and vincristine.	59% at 3 years
	M2	3	39.68 Gy/1.24 Gy BID	22.32 Gy/1.24 Gy BID		
	M3	22	39.68 Gy/1.24 Gy BID	22.32 Gy/1.24 Gy BID		
SJMB03 [23]	M0	2	36 Gy/1.8 Gy	Boost to TB up to 55.8–59.4 Gy Boost to metastatic sites > 0.5 cm up to 50.4–59.4 Gy	Four cycles of cyclophosphamide (4 g/m^2^), cisplatin (75 mg/m^2^), and vincristine (1.0 mg/m^2^) with autologous stem cell rescue.	5 years PFS 56.7% 5 years OS 69.5%
	M1	17	36 Gy/1.8 Gy			
	M2	28	39.6 Gy/1.8 Gy			
	M3	56	39.6 Gy/1.8 Gy			
ACNS0332 [37]	M0	72	36 Gy/1.8 Gy	Boost up to 55.8 Gy	Randomization: carboplatin-vincristine versus vincristine during radiotherapy. All patients received six courses of post-radiation cyclophosphamide-cisplatin-vincristine. A second randomization to isotretinoin maintenance was discontinued in 2015 for futility.	5-year PFS 62.9% 5-year OS 73.4% For group 3 M+ patients, 5-year PFS was 64.3% with carboplatin versus 40.3% without
	M1	33	36 Gy/1.8 Gy	Boost up to 55.8 Gy		
	M2	41	36 Gy/1.8 Gy	Boost up to 55.8 Gy		
	M3	115	36 Gy/1.8 Gy	Boost up to 55.8 Gy		
PNET HR + 5	M0	14	23.4 Gy/1.8 Gy	Boost up to 54 Gy/1.8 Gy	Two courses of etoposide-carboplatin followed by two courses of thiotepa (600 mg/m^2^) with PBSC rescue, 3 weeks apart. Patients then proceeded to radiotherapy—maintenance with six courses of temozolomide.	5-year PFS and OS 76%
	M1	3	36 Gy/1.8 Gy	Boost up to 54 Gy/1.8 Gy		
	M2/3	34	36 Gy/1.8 Gy	Boost up to 54 Gy/1.8 Gy		

PFS: progression free survival; OS: overall survival; Gy: Gray; TB: tumor bed; MTX: methotrexate; XRT: radiotherapy; CR: complete response; HDMTX: High dose methotrexate.

The evolution of systemic therapies for high-risk patients has been more complex, and there are still numerous controversies concerning optimal management. North American protocols essentially use upfront radiotherapy followed by adjuvant chemotherapy [10]. By contrast, most European protocols use pre-radiotherapy chemotherapy [27,28]. 

In North America, one major focus has been the administration of chemotherapy concurrently with craniospinal radiotherapy. The phase II study POG 9631 enrolled 47 patients with high-risk medulloblastoma, including 35 with metastatic disease. The patients received craniospinal radiotherapy at a dose of 36 Gy with concomitant oral etoposide [49]. The trial was amended after the inclusion of the first 12 patients, as the daily dose of 50 mg/m^2^ was associated with a high rate of dysphagia/esophagitis. The dose was therefore reduced to 35 mg/m^2^. The design of the trial was planning a comparison with the response rate observed in the POG 9031 protocol (radiation therapy followed by adjuvant chemotherapy) [48]. As the 2- and 5-year progression-free and overall survival rates were not different between this trial and POG 9031, no further development was planned. Another phase I/II study evaluated the feasibility of administering carboplatin as a radiosensitizer during CSI to high-risk medulloblastoma patients. The patients were to receive daily carboplatin at a dose of 35 mg/m^2^, 1 to 4 h before each session of radiotherapy and a weekly administration of vincristine. The patients initially received 15 doses of carboplatin (3 weeks). The number of doses of carboplatin was progressively increased to 20, 25, and 30, and the dose of carboplatin was increased in increments of 5 mg/m^2^/dose up to 50 mg/m^2^/dose. Following CSI, patients received six courses of adjuvant chemotherapy. The maximum recommended phase II dose of carboplatin was 35 mg/m^2^ for 30 sessions, with myelosuppression the dose limiting toxicity [50]. Subsequently, a randomized trial compared this chemo-radiation design with CSI and weekly vincristine followed in both arms by the administration of six courses of adjuvant cyclophosphamide, cisplatin, and vincristine. The trial included a second randomization with or without maintenance isotretinoin after the completion of chemotherapy [37]. Overall, this trial did not show a significant survival benefit associated with the administration of carboplatin during CSI, with a 5-year overall survival of 77.6% with carboplatin versus 68.8% for the control arm (*p* = 0.28). However, subgroup analyses identified an effect of carboplatin on outcome exclusively in group 3 patients, with a 5-year event-free survival of 73.2% with carboplatin versus 53.7% without carboplatin (*p* = 0.047). For patients with metastatic group 3 medulloblastoma, 5-year PFS was 64.3% with carboplatin versus 40.3% without. The randomization to isotretinoin was discontinued following an interim futility analysis that demonstrated that the addition of isotretinoin was unlikely to show significant event-free survival differences. The result of this trial is setting a new standard for group 3 high-risk medulloblastoma patients. However, this requires a timely and reliable subgrouping of tumor samples prior to the initiation of radiotherapy, which is currently unrealistic in many centers worldwide. 

Three North American initiatives also explored the use of high-dose chemotherapy given after CSI. Protocol ACNS99702 included one course of chemotherapy prior to CSI to allow peripheral blood stem cell (PBSC) collection to support subsequent consolidation therapy [55]. Following CSI, patients received three courses of consolidation with high-dose chemotherapy and PBSC rescue. Courses 1 and 3 included the administration of high-dose thiotepa. The study was closed early due to an unacceptable rate of veno-occlusive disease of the liver. SJMB96 and SJMB03 both used a combination of cisplatin, high-dose cyclophosphamide, and vincristine with autologous PBSC transplant for four courses after dose-adapted CSI (23.4 Gy in average-risk patients, 36 Gy in high-risk patients) [23,33]. There was no case of veno-occlusive disease reported and the 5-year event-free survival of high-risk patients in SJMB03 was 56.7%, with large differences between subgroups (25% for SHH medulloblastoma patients, 100% for WNT medulloblastoma patients). Although this experience did not suggest a survival advantage with this approach [56], it allowed a better identification of high-risk patients such as patients with metastatic disease or subtype III or medulloblastoma with MYC amplification.

In Europe, a unicentric pilot study from the Milan group reported the results of a protocol combining HART (hyperfractionated accelerated radiotherapy) with intensive sequential chemotherapy [53]. Prior to HART, patients received a 7-week course of chemotherapy combining high-dose methotrexate, vincristine, etoposide, cyclophosphamide, and carboplatin. Following HART, patients in complete response received maintenance chemotherapy with vincristine and CCNU, while incomplete responders received two courses of high-dose thiotepa with PBCS rescue. The toxicity of this approach was manageable, and the 3- and 5-year progression-free survival was 80% and 72%, respectively. However, a UK attempt to reproduce these results in a multicenter setting failed and showed a lower complete response rate (16/34 patients versus 30/33 patient in the Milan experience) as well as a lower 3-year event-free survival of 56% [54]. 

The French group conducted a phase II trial that included two courses of postoperative induction chemotherapy with etoposide and carboplatin, followed by two courses of high-dose thiotepa (600 mg/m^2^) with hematological stem cell support [28]. The PBSC collection was planned after the first or second course of etoposide and carboplatin. Patients with metastatic medulloblastoma then received a 36 Gy dose of CSI, whereas high-risk non-metastatic patients (with anaplasia or postoperative residue >1.5 cm^2^) received a lower dose of CSI (23.4 Gy). The patients then received maintenance chemotherapy with six courses of oral temozolomide. The trial accrued 51 patients (37 with metastatic disease), and 33 completed the whole scheduled program. Twenty-five patients achieved a complete response before radiotherapy, thirteen a partial response, eight had stable disease, and three had progressive disease (two were not assessed). Twelve patients were unable to receive the maintenance chemotherapy due to persistent thrombocytopenia. At a median follow-up of 7.1 years, the 3- and 5-year progression-free survival rates were 78% and 76%, and the 3- and 5-year overall survival rates were 84% and 76%, respectively. All WNT and group 4 medulloblastoma patients were alive at the time of publication. 

The German group reported the result of the multicenter HIT2000 trial for metastatic medulloblastoma patients. This trial included a pre-radiation induction chemotherapy followed by CSI using hyperfractionation and maintenance chemotherapy [27]. The induction used two cycles of the SKK-HIT protocol, i.e., a combination of intravenous cyclophosphamide, vincristine, high-dose methotrexate, carboplatin, and etoposide, and concomitant intraventricular methotrexate. The maintenance consisted of four courses of cisplatin, CCNU, and vincristine. In this cohort of 123 eligible patients, the 5-year event-free survival and overall survival rates were 62% and 74%. Non-responders after one cycle of induction chemotherapy had a significantly poorer outcome. There was no clear association between patients with non-responding tumors and MYCC/MYCN status, histologic subtype, or molecular subtype. Subgrouping was available for 69 patients. The study did not detect an overall difference among the four subgroups.

Overall, high-risk protocols for older patients have contributed to improve knowledge. In particular, the interpretation of these results in the context of medulloblastoma subgrouping has shown that not all high-risk/metastatic patients are similar, opening the prospect of risk-adapted protocols for high-risk patients [57]. The SIOPE group has already opened a de-escalation trial for metastatic WNT patients who are receiving a reduced dose of CSI (23.4 Gy) followed by adjuvant chemotherapy [35]. The same group has also opened a specific stratum for patients with TP53-mutated SHH medulloblastoma, as these patients have been identified to have a particularly poor prognosis, with a significant proportion of these tumors occurring in the context of Li–Fraumeni syndrome [58,59]. The patients receive two cycles of HIT-SKK-like induction, followed by focal radiotherapy when the disease is localized or CSI in the context of metastatic disease. A “light” maintenance then consists of the administration of vinblastine for 24 weeks. The SIOP-Europe HR-MB trial for high-risk patients is currently open and enrolling. The design of this trial is complex, and the primary objective is to compare the outcome of patients treated according to three different modalities: daily conventional fractionation of radiotherapy, HART radiotherapy, and high-dose chemotherapy followed by conventional radiotherapy. The trial will also evaluate the benefits of two different maintenance chemotherapy regimens. This trial is programmed to recruit over 800 patients in 16 countries across Europe [52]. 

## 6. Radiation-Sparing Approaches in Younger Children (Table 3)

The devastating consequences of craniospinal radiotherapy in infants and young children have been extensively described, and there is a consensus that this treatment modality should be avoided, due to its lifelong consequences. Walter et al. described the neurocognitive outcome of 14 infants and young children successfully treated with adjuvant or salvage radiotherapy [60]. The median age of these children at the time of radiotherapy was 3.0 years old. With a follow-up of 4.8 years, the median IQ of these survivors was 62 (range 44–86), with an average annual decline of IQ of −3.9. However, while the benefit of this radiation sparing approach in terms of neurocognitive outcome has been demonstrated, the results of treatments in young children with medulloblastoma treated with postoperative chemotherapy only are inferior to those reported in older children treated with surgery, craniospinal irradiation, and chemotherapy [12]. 

**Table 3 diagnostics-13-03680-t003:** Infant studies.

Infant MB	Risk Category	Number	Radiotherapy/ Fractionation	Chemotherapy Regimen	Outcome DMB/MBEN	5-Year PFS Others
Protocol						
COG 9934 [61]	M0	74 (39 DMB)	18 to 23.4 Gy (PF) and boost up to 50.4–54 Gy/1.8 Gy TB	Four courses of induction chemotherapy (cyclophosphamide, vincristine, cisplatin, and etoposide) followed by focal PF radiotherapy and four cycles of cyclophosphamide and vincristine followed by oral etoposide.	4-year PFS 58%	4-year PFS 23%
SKK91 [26]	M0/1/2/3	43 (20 DMB/MBEN)	None	SKK-HIT: three cycles of alternating courses of (1) cyclophosphamide-vincristine, (2 and 3) high-dose MTX and vincristine, and (4) carboplatin-etoposide. Patients received intraventricular MTX during each course.	7-year PFS 85%	7-year PFS 34%
					7-year OS 95%	7-year OS 41%
SKK 2000 [14,62]	M0	87 (42 DMB/MBEN)	After 2006, patients with classic or LCA MB in incomplete remission received 54 Gy focal radiotherapy	Three cycles of HIT-SKK with intraventricular MTX.	5-year PFS 93% 5-year OS 100% 5-year CSI-free OS 91%	5-year PFS 37% 5-year OS 62%. No impact of XRT
Head Start 3 [16]	M0/1/2/3	83 56 ≤3 years old 27 DMB/MBEN	Focal if local residual (55.8 Gy), CSI if residual dissemination (23.4 Gy)	Five cycles of induction chemotherapy (three cycles of cisplatin, vincristine, cyclophosphamide, etoposide, and high-dose MTX, alternating with two cycles of cyclophosphamide, etoposide, vincristine, and temozolomide; consolidation with high-dose carboplatin, thiotepa, and etoposide, and stem cell transplant.	5-year PFS and OS 89% 5-year radiation-free 78%	5-year PFS classic MB 26% 5-year PFS LCA MB 38%
Baroni [63]	≈	23 (16 SHH)	PF radiotherapy to 54 Gy/1.8 Gy	Five cycles of cisplatin, vincristine, etoposide, cyclophosphamide, and MTX (only for metastatic patients), followed by focal radiotherapy (children > 18 months) with concomitant temozolomide, then maintenance with carboplatin, vincristine, cyclophosphamide, and etoposide (four 56-day cycles).	5-year PFS SHHδ 100% 5-year PFS SHHꞵ 56%	5-year PFS Group 3 50%
	M+	4 (1 SHH)				
SJYC07 [64]	Low risk	9	None	Four cycles of high-dose MTX, vincristine, cisplatin, and cyclophosphamide. High-risk patients also received vinblastine (five doses/cycle)—consolidation is risk-specific (low-risk carboplatin-etoposide-cyclophosphamide ×2; intermediate: radiotherapy; high-risk: either CSI or IV topotecan-cyclophosphamide); all risk groups then receive six cycles of oral maintenance with cyclophosphamide, topotecan, and erlotinib.	5-year PFS M + SHH-II 100% 5-year PFS M0 SHH-II 72% 5-year PFS M + SHH-I 25% 5-year PFS M0 SHH-II 28%	5-year PFS intermediate-risk 24.6% 5-year PFS high-risk 16.7%
	Intermediate risk	6	Focal XRT to TB			
	High risk	27	CSI (optional)			
ACNS1221 [65]	M0	25 DMB	No	SKK-HIT without intraventricular MTX.	2-year PFS SHH1 30% 2-year PFS SHH2 66.7%	

MB: medulloblastoma; DMB: desmoplastic medulloblastoma; MBEN: medulloblastoma with extensive nodularity; LCA: large cell/anaplastic medulloblastoma; PFS: progression-free survival; OS: overall survival; Gy: Gray; PF: posterior fossa; TB: tumor bed; CSI: craniospinal; XRT: radiotherapy; MTX: methotrexate. SHH: Sonic hedgehog; SKK HIT: Säuglinge und Kleinkinder mit Hirntumoren.

Evidence that young children can be successfully treated with surgery followed by chemotherapy without radiotherapy stems from the so-called Baby POG1 protocol [13]. In this trial, children less than 3 years of age were treated with a four-drug regimen (courses of vincristine-cyclophosphamide and etoposide-cisplatin). Medulloblastoma patients were to receive CSI at the age of 3. However, the parents of 13 children who had no evidence of residual tumor at the end of the chemotherapy treatment declined treatment with radiation. Eleven patients were disease-free at a median of 1 year following the completion of chemotherapy. Based on this evidence and with increasing awareness of the devastating effects of CSI in young patients, subsequent protocols in North America and in Europe avoided radiation as a first-line treatment. However, the results of these trials were disappointing. In the CCG9921 trial that randomized two regimens of induction chemotherapy (vincristine, cisplatin, cyclophosphamide, and etoposide versus vincristine, carboplatin, ifosfamide, and etoposide), only 20 of 92 infants and young children with medulloblastoma were alive radiation-free at 5 years [66]. A major breakthrough was the publication of a study by Rutkowski et al. in 2005, who described 43 infants and young children treated with an intensive regimen involving three cycles of multiagent chemotherapy (cyclophosphamide, vincristine, high-dose methotrexate, carboplatin, and etoposide) and the administration of intraventricular methotrexate [26]. Children who had a complete resection and no metastatic disease had 5-year progression-free and overall survival rates of 82 ± 9% and 93 ± 6%. Children with localized disease and incomplete resection and children with metastatic disease had a 5-year progression-free and overall survival of 50 ± 13% and 56 ± 14%, and 33 ± 14% and 38 ± 15%, respectively. Importantly, this trial identified desmoplastic histology as a major prognostic factor, and the 5-year event-free survival of 20 patients with desmoplastic histology was 85% versus 34% for patients with classic histology. The role of histology was subsequently confirmed in a large meta-analysis of European and North American trials. Interestingly, this meta-analysis showed that national protocols were independent risk factors for event-free and overall survival, with higher survival rates in the German and North American (HeadStart) protocols, while the UK protocol had the lowest survival rates. A subsequent trial of the German group, HIT-SKK 2000, confirmed the high survival rate of patients with desmoplastic medulloblastoma (DMB) and medulloblastoma with extensive nodularity (MBEN). This protocol also showed that all DMB/MBEN assessed by DNA methylation profiling belonged to the subgroup of infantile Sonic Hedgehog (SHH) medulloblastoma [14]. In infants and young children with non-SHH medulloblastoma, the 5-year progression-free survival was 37% and the CSI-free survival was 39%. The addition of local radiotherapy in this subgroup of patients did not improve survival. 

As intraventricular chemotherapy was potentially associated with a risk of encephalopathy, and due to the lack of randomized evidence of its benefit, two North American trials were developed simultaneously. Their objective was to reproduce the results of the HIT-SKK 91 and 2000 protocols without the use of intraventricular chemotherapy. ACNS1221 was a multicenter COG trial for patients less than 4 years old with non-metastatic DMB/MBEN medulloblastoma [65]. This trial closed early, due to a high rate of relapses (13 out of 25 patients), with a 2-year event-free survival rate of 52%. Interestingly, none of the patients with MBEN and/or less than 12 months of age experienced a relapse. SJYC07 was a risk-adapted trial for children up to the age of 5, depending on clinical and histological criteria [64]. This trial also omitted intraventricular methotrexate and similarly showed a high rate of relapses in DMB/MBEN patients. The outcome of patients without metastatic disease and classic histology was also disappointing despite the use of focal radiotherapy (24.7% 5-year event-free survival). By subgroups, patients in group 3 had a 5-year event-free survival of 8.3% versus 13.3% in group 4 and 51.1% in the SHH subgroup. 

Another radiation-sparing approach for this vulnerable group of patients has been the use of high-dose chemotherapy with bone marrow or peripheral stem cell transplant. HeadStart protocols I, II, and III were pivotal trials that demonstrated the feasibility of high-dose chemotherapy as a consolidation following an intensive induction regimen [16,29,67]. However, the lack of central histology review and more recently the lack of subgrouping in HeadStart III make the interpretation of these trials challenging. In HeadStart III, patients were initially treated with five courses of induction. Courses 1, 3, and 5 included vincristine, cyclophosphamide, cisplatin, etoposide, and high-dose methotrexate, whereas courses 2 and 4 included cyclophosphamide, vincristine, oral etoposide, and temozolomide. The consolidation consisted of one course of high, dose chemotherapy with carboplatin, etoposide, and thiotepa, followed by stem cell rescue. The 5-year radiation-free survival for patients with DMB/MBEN medulloblastoma was 78% compared to 21% for patients with other histologies [16]. 

Over the past 20 years, sequential high-dose chemotherapy has been introduced in the management of brain tumor in infants. Protocol CCG99703 was a phase I study that demonstrated the feasibility of three courses of high-dose thiotepa containing chemotherapy with a maximum dose of thiotepa of 20 mg/kg per course [68]. This trial enrolled 36 infants and young children with medulloblastoma and reported a 5-year EFS of 60%. However, the interpretation of these results is flawed as this trial was not designed to investigate the role of radiotherapy, and data on additional radiotherapy treatments were not collected. A retrospective review of patients treated off-protocol with the same approach showed a 5-year progression-free survival (PFS) and overall survival (OS) of 69.6% and 76.1%, respectively [69]. Patients with SHH medulloblastoma had a 5-year PFS of 86% compared to 49% for group 3 patients, and the 5-year radiation free survival for group 3 patients was 46.4%. Such results are promising for group 3 patients, who account for the largest subgroup of medulloblastoma in this age group along with the SHH subgroup. The ACNS0334 protocol is comparing two different three-course induction regimens with and without high-dose methotrexate, followed by three courses of high-dose chemotherapy with carboplatin and thiotepa, and stem cell rescue in children <3 years of age with high-risk medulloblastoma (metastatic disease and/or postoperative residual >1.5 cm^2^). Early reports have suggested a higher response rate and better event-free and overall survival in the group of patients treated with high-dose methotrexate [70]. The final results of this trial are awaited. 

Overall, current options for infant and young children with medulloblastoma include a “HIT-SKK” approach for patients with DMB/MBEN histology or SHH subgrouping. The evidence that SHH includes different subtypes with different behaviors [64,71] has not translated in the design of clinical trials. The benefit of high-dose chemotherapy in this category of patients needs to be investigated, and an international randomized study is planned to compare the HIT/SKK protocol and a high-dose chemotherapy approach with survival and neurocognitive outcomes as the primary endpoints. For infants and young children with non-SHH medulloblastoma, the results of radiation-sparing protocols are still disappointing, with a 5-year EFS ranging between 20 and 35%. The results of protocols using sequential high-dose chemotherapy are encouraging, and this approach needs to be studied further [68,69]. This is the objective of the HeadStart IV trial (NCT02875314) that is comparing one consolidation versus three consolidations in this group of patients. Finally, the management of infants and young children with metastatic medulloblastoma is still a major challenge. While metastatic disease does not seem to affect the outcome of patients with DMB/MBEN histology, the radiation-free survival of metastatic patients with classic histology is dismal. In a review of 74 patients with non-WNT, non-SHH metastatic medulloblastoma treated with a radiation-sparing approach including high-dose chemotherapy, Mynarek reported a 5-year CSI-free survival of 8% [72]. New strategies are definitely needed for this group of patients. 

## 7. Chemotherapy as a Salvage Treatment

Very few prospective trials have been conducted in the recurrent setting, and there is no consensus on the optimal management of recurrent medulloblastoma. For young patients initially treated with a radiation-sparing approach, the use of craniospinal irradiation is a logical option. The choice of the dose of CSI is still a matter of debate. A recent review has shown that for patients who received a reduced dose of CSI (<35 Gy), the addition of systemic chemotherapy was associated with improved progression-free survival compared to CSI alone (55% 3-year survival versus 38%; *p* = 0.007) [73].

For medulloblastoma patients who relapse after a CSI-containing regimen, studies conducted in the 1990s have suggested a survival benefit with high-dose chemotherapy followed by bone marrow or peripheral stem cell transplant [74,75,76]. As a result, this approach has become a standard of care that is still used in some institutions. However, the role of high-dose chemotherapy in this setting has been eventually questioned by cooperative groups or in institutional reviews [77,78,79,80]. The recent evidence that medulloblastoma subgrouping influences survival both at the time of diagnosis and after recurrence suggests that most long-term survivors of early reports on high-dose chemotherapy were probably group 4 patients [81].

There is an increasing interest in metronomic, low-dose chemotherapy for patients with recurrent medulloblastoma. The MEMMAT protocol has provided promising results in selected patients [82,83]. This protocol combines chemotherapy agents (cyclophosphamide, etoposide, intraventricular etoposide, and cytarabine), bevacizumab and oral thalidomide, fenofibrate and celecoxib. In a phase II trial of 40 patients with recurrent medulloblastoma from several institutions in Europe and North America treated with this approach, Peyrl et al. reported six complete responses (15.0%), nine partial responses (22.5%), and five stable diseases (12.5%) The 5-year survival rate of patients who had no evidence of progression at 12 months was 66.7% [84]. 

A phase II trial from the COG compared the outcome of patients with recurrent medulloblastoma treated with temozolomide and irinotecan with or without bevacizumab [85]. This trial showed a significantly longer progression-free survival with the addition of bevacizumab (9 versus 6 months). This trial did not analyze outcomes according to subgrouping, and the specific impact of the addition of bevacizumab in this context remains unknown.

## 8. Medulloblastoma Management in Countries with Limited Resources

Prospective clinical trials in the LMIC setting are few. Most information relies on retrospective studies, limiting a critical evaluation of the role of systemic chemotherapy in this context. In addition, LMICs face significant challenges such as limited access to appropriate facilities (imaging, neurosurgical, pathology, and radiotherapy), a lack of universal health care coverage, a high rate of abandonment, and a high incidence of treatment refusal [86,87]. When it relates to the current WHO classification, gene sequencing and methylation are unaffordable in most institutions, and the subgrouping—when it is done—most often relies on immunohistochemical staining. However, some experiences are noteworthy. Gupta et al. (Tata Memorial Hospital, India) reported the feasibility and efficacy of the addition of carboplatin during craniospinal irradiation in a series of 97 high-risk medulloblastoma patients, including 45 metastatic patients, with a 5-year EFS and OS of 60.2% and 62.1%, respectively [88]. Baroni et al. (Argentina) reported the outcome of 29 young children treated with a craniospinal irradiation-sparing strategy [63]. The children were treated with five cycles of induction chemotherapy that consisted of a combination of cisplatin, etoposide, cyclophosphamide, and vincristine. Patients with metastatic disease or high-risk features (postoperative residue > 1.5 cm^2^) also received high-dose methotrexate. Patients older than 18 months received focal radiotherapy to the posterior fossa, with concomitant daily temozolomide. Patients younger than 18 months received maintenance chemotherapy until the age of 18 months. Maintenance chemotherapy consisted of four 56-day cycles of carboplatin, vincristine, cyclophosphamide, and oral etoposide. In this experience, the 5-year PFS was 70.4% and the 5-year OS was 77.8%. These experiences suggest that the implementation of modern strategies is possible in LMICs. Considering the relative success rate of MEMMAT as a salvage treatment [84], the affordably of the drugs, and the possibility to manage patients in the outpatient setting, MEMMAT would be an interesting option for patients with medulloblastoma in LMICs in the first-line setting. However, Ommaya reservoir insertion is not routine in many LMICs. Several additional factors that are beyond the scope of this review impact on the management of medulloblastoma in LMICs, including late diagnoses, late referrals, and a lack of multidisciplinary care. 

## 9. The Future

While the benefit of chemotherapy in the management of patients with medulloblastoma has been questioned for decades, there is now a worldwide consensus that adjuvant chemotherapy is an important component of medulloblastoma protocols. The introduction of subgrouping in new protocols opens new perspectives, with treatments based on specific risk stratifications. There is still an urgent need to explore new therapies, in particular for high-risk and very high-risk patients. This includes patients with group 3 metastatic disease, patients with Myc-amplified medulloblastoma, patients with SHH TP53-mutated medulloblastoma, and infants and young children with non-SHH metastatic medulloblastoma. For these patients, intensive and high-dose chemotherapy protocols have failed to demonstrate a benefit, and there is an urgent need to identify alternatives to a standard chemoradiation approach.

## Data Availability

No new data were created or analyzed in this study. Data sharing is not applicable to this article.

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
