# Peer review of "Evolution of Systemic Therapy in Medulloblastoma Including Irradiation-Sparing Approaches"

_diagnostics, 2023, doi:10.3390/diagnostics13243680_

Round 1

Reviewer 1 Report

Comments and Suggestions for Authors

Dear Dr. Mushtaq,

I am sorry to judge that we must reject your manuscript of a review titled “Evolution of systemic therapy in medulloblastoma' including irradiation-sparing approaches” as the current form. I understand and admire your huge effort to collect the data of clinical studies for children with medulloblastoma.

However, your way of integration using the huge data is not sufficient in quality. Most of readers could not catch what you would like to discuss in this review article in the current form. Because there are too many things to be corrected or improved, let me just list up several advices, which are vital to improve your manuscript, in the following part.

In addition, the contents of your article are not appropriate for our journal “Diagnostics”, and please consider re-submission to another journal handling treatment of pediatric oncology or neuro-oncology.

1. In the Title, you suggest that “irradiation-sparing approaches” is one of the main topics. Whereas, in the Introduction, you emphasized “specific focus” on the management in LMIC. There is no consistency from the initial part. Moreover, you only wrote sixteen lines for the latter topic with “specific focus”. You should design the total construction of the article at first, then, start to construct your logic carefully.

2. You should provide more detail treatment information about the clinical studies. You mainly describe name of study groups, protocol code, name of drugs (without doses and schedules) and simple outcome. I do not believe that most of the readers could not follow what you would like elaborate about. To discuss the treatment evolution, you have to describe the precise comparison among sequential studies (i.e., what was changed and what was the outcome because of it). Maybe, you should use Tables and Figures to summarize such information.

3. Your co-author, Dr. Bouffet, is a really great neuro-oncologist, who have written a number of excellent academic papers. It is hardly believable that he agreed with the quality of your current manuscript for submission. Please ask him to give you a strict proofreading prior to the next submission.

Author Response

Dear Editor,

We are grateful for considering our manuscript for publication and would like to thank the reviewers for their valuable comments and suggestions.

Reviewer 1

We really appreciate the comment of reviewer 1 regarding the amount of work involved in this extensive review. We are sorry to see that the result did not live up to his expectations. We see that Reviewer 1 gave very low scores for this work (3 one star, one 2 stars, and one 3 stars) and we did our best to take into account his/her comments in order to improve the manuscript.

Regarding the suggestions

  • In addition, the contents of your article are not appropriate for our journal “Diagnostics”, and please consider re-submission to another journal handling treatment of pediatric oncology or neuro-oncology.

 We do agree that the title of the manuscript may not seem to be appropriate for the journal “Diagnostics”. However, this work was an invited manuscript, part of a special series on pediatric brain tumors, and we did our best to meet the expectations of the guest editor.

  • In the Title, you suggest that “irradiation-sparing approaches” is one of the main topics. Whereas, in the Introduction, you emphasized “specific focus” on the management in LMIC. There is no consistency from the initial part. Moreover, you only wrote sixteen lines for the latter topic with “specific focus”. You should design the total construction of the article at first, then, start to construct your logic carefully.

Thank you for this important point. In the introduction, we removed any comment pertaining to LMIC. Literature regarding medulloblastoma in LMIC is complex and challenging, and this would deserve a special manuscript. However, we kept a small section toward the end to the manuscript, as it seems important to acknowledge innovative studies conducted by some teams in LMIC.

  • You should provide more detail treatment information about the clinical studies. You mainly describe name of study groups, protocol code, name of drugs (without doses and schedules) and simple outcome. I do not believe that most of the readers could not follow what you would like elaborate about. To discuss the treatment evolution, you have to describe the precise comparison among sequential studies (i.e., what was changed and what was the outcome because of it). Maybe, you should use Tables and Figures to summarize such information.

Thank you for pointing out this issue. We did not know whether the addition of tables would strengthen the manuscript. We have now added 3 tables which summarize recent clinical trials or important studies, one for average risk patients, one for high risk patients, and one for patients treated with a radiation (CSI) sparing approach. As there is always room for improvement, we would appreciate any comment that could improve the content of these tables. 

Reviewer 2 Report

Comments and Suggestions for Authors

This review describes changes that have occurred in the management of pediatric medulloblastoma in the context of our greater understanding of the molecular and histologic subtypes observed over the years. This review describes clinical trials and their outcomes but does not go into mechanistic discussions of the therapies used. This is not necessarily a shortcoming of this review, as that was not its goal; however, references that speak to this would have been good.

Comments on the Quality of English Language

The manuscript needs to be edited to correct a number of minor grammatical and syntax errors. A few examples:

Line 30 “tumor is the” should be “in the”

Line 52 “… can find found”

Line 75 “… and are” should be “and we are”

Lines 96-97 should be rewritten

There are often errors in pluralization for example in line 170 ”in both arm”, often the authors say 5 years progression instead of 5 year, etc.

Author Response

Reviewer 2

We are extremely grateful for the comments of reviewer 2 and his positive suggestions.

  • This review describes changes that have occurred in the management of pediatric medulloblastoma in the context of our greater understanding of the molecular and histologic subtypes observed over the years. This review describes clinical trials and their outcomes but does not go into mechanistic discussions of the therapies used. This is not necessarily a shortcoming of this review, as that was not its goal; however, references that speak to this would have been good.

Thank you for this important comment. We have added some references on mechanistic discussions of therapies used, particularly SHH inhibitors.

  • The manuscript needs to be edited to correct a number of minor grammatical and syntax errors. A few examples: Line 30 “tumor is the” should be “in the”; Line 52 “… can find found”; Line 75 “… and are” should be “and we are” Lines 96-97 should be rewritten. There are often errors in pluralization for example in line 170 ”in both arm”, often the authors say 5 years progression instead of 5 year, etc.

We have reviewed the text and correct these grammatical errors.

Reviewer 3 Report

Comments and Suggestions for Authors

The manuscript by Mushtaq and collaborators discusses an extremely relevant topic for improving the diagnosis and subsequent treatment of medulloblastoma. However, I believe that the manuscript would improve greatly if the authors presented medulloblastoma with a translational view between basic and clinical research findings. The following are suggestions for improving the manuscript.

1 - The authors do not describe in the introduction how medulloblastoma is classified according to the World Health Organization (WHO). The authors should describe this classification and correlate it with molecular markers normally found in clinical practice.

2 – Authors must describe the articles that demonstrate for the first time the deregulation of the Wnt and Shh pathways in medulloblastoma and the articles that demonstrate for the first time the action of inhibitors of these pathways. In lines 142, 143 the authors draw attention to the discovery of Shh inhibitors but do not provide a reference or how these inhibitors were discovered.

3 – the manuscript has information that was not referenced, such as:

Lines 79-84

As far as chemotherapy agents are concerned, protocols for older children in the upfront setting have shown little changes and include a limited number of agents, mostly cisplatin, cyclophosphamide, vincristine, and lomustine in North America, whereas European protocols also use etoposide, carboplatin, ifosfamide and high dose methotrexate. In younger children, in addition to these drugs, thiotepa is often part of high dose chemotherapy regimens. Protocols for young children and in some cooperative groups for metastatic patients use repeated injections of intraventricular chemotherapy.

Lines 116 a 118

In this protocol patients with WNT and SHH medulloblastoma receive 4 cycles of cyclophosphamide-cisplatin and vincristine. Post-pubertal patients with SHH medulloblastoma receive a Vismodegib maintenance. Patients with non SHH-Non WNT medulloblastoma receive 4 to 7 cycles of chemotherapy.

4 – The authors should create a table with all clinical trials described in the text.

5 – The authors describe several clinical trials using combinations of radiotherapy and chemotherapy. Would it not be possible to carry out a meta-analysis of this data?

Comments on the Quality of English Language

The manuscript writing needs to improve. Some information is difficult to understand due to grammatical errors

Author Response

Reviewer 3

We thank Reviewer 3 for the kind comments on our manuscript and the important suggestions made.

  • The authors do not describe in the introduction how medulloblastoma is classified according to the World Health Organization (WHO). The authors should describe this classification and correlate it with molecular markers normally found in clinical practice.

We have now added a section on the subgrouping of medulloblastoma and the techniques used to achieve a reliable classification.

  • Authors must describe the articles that demonstrate for the first time the deregulation of the Wnt and Shh pathways in medulloblastoma and the articles that demonstrate for the first time the action of inhibitors of these pathways. In lines 142, 143 the authors draw attention to the discovery of Shh inhibitors but do not provide a reference or how these inhibitors were discovered. Done for SHH, not for WNT

We have added some details on the background and discovery of SHH inhibitors and added important references regarding these agents. However, we did not expand on WNT, as there is no WNT inhibitor currently used or in development in medulloblastoma.

  • the manuscript has information that was not referenced, such as:

Lines 79-84: As far as chemotherapy agents are concerned, protocols for older children in the upfront setting have shown little changes and include a limited number of agents, mostly cisplatin, cyclophosphamide, vincristine, and lomustine in North America, whereas European protocols also use etoposide, carboplatin, ifosfamide and high dose methotrexate. In younger children, in addition to these drugs, thiotepa is often part of high dose chemotherapy regimens. Protocols for young children and in some cooperative groups for metastatic patients use repeated injections of intraventricular chemotherapy.

We have added specific references to support these statements.

Lines 116 a 118: In this protocol patients with WNT and SHH medulloblastoma receive 4 cycles of cyclophosphamide-cisplatin and vincristine. Post-pubertal patients with SHH medulloblastoma receive a Vismodegib maintenance. Patients with non SHH-Non WNT medulloblastoma receive 4 to 7 cycles of chemotherapy.

 The reviewer is right, there is no report on this trial so far. The NCT number give access to some information regarding this trial. We have added the URL to the NCT number.

  • The authors should create a table with all clinical trials described in the text.

There are now 3 table summarizing clinical trials (average risk patients, high risk patients and infants)

  • The authors describe several clinical trials using combinations of radiotherapy and chemotherapy. Would it not be possible to carry out a meta-analysis of this data?

Thank you for this suggestion. Although this is not the scope of this review, we think that this is an important  and will consider a meta-analysis in the future.

Reviewer 4 Report

Comments and Suggestions for Authors

This is a well-written article that provides a thorough review of the developmental history of chemotherapy in medulloblastoma with a global perspective. 

Major suggestions: 

Would this paper about cancer treatment be more suitable to a cancer journal rather than a general dianostic journal? The authors may discuss this issue with editors and the publisher. 

The authors may explain why they chose to focus on irradiation-sparing approaches. Was this due to limited access to radiation therapy in several settings? Would many families chose no radiation therapy because of unaffordable cost or the difficulties of survivorship (e.g. raising a surviving child/adolescent/adult with neurocognitive impairments) in LMICs? Backgrounds that facilitated this review may be discussed. 

Abstract 

Line 17: Define "high-dose" radiotherapy. 

Lines 19-20: Does "increasing complex condition" mean medulloblastoma? If so, the word "condition" may be changed into "disease". If the authors want to raise other issues, e.g. survivorship of MB, more information should be provided to explain the complexity of MB management. 

1. Introduction 

More diagnostic aspects related to MB treatment should be discussed. Can the molecular subgroups be identified by pathologists in LMIC settings? Are there regional collabrative networks that may help to establish the molecular diagnosis (or a diagnosis that is compatible with the WHO 2021 scheme)? 

Would the authors consider the recent discovery of sub-subgroups of MBs (e.g. alpha to delta of SHH MB) is clinically relevant? Please provide your point of view with citation(s). 

2. Evolution 

Although it's a paper of systemic therapy in MB, the importance of neurosurgery and radiation therapy should still be briefly discussed here or in earlier paragraphs. The impact of gross total resection and near-total resection should be reviewed. Studies from LMICs would of particular value. 

The authors may also discuss about - 

why young and old children with MB have been treated differently? 

why are there different age cutoffs in MB protocols and what are the commonly used cutoffs in LMICs? 

what are the patterns of radiation therapy for children with MB in LMICs? 

Section #6 nicely reviewed the discovery of systemic therapy contributed by studies in LMICs. May the authors also address other issues in resource-limited settings, e.g., abandonment, treatment refusal, misdiagnosis, patient access to diagnosis and/or follow-ups, and toxic deaths? 

Minor points: 

Which author(s) is/are from Affiliation (4)? 

The affiliation of the corresponding author was not provided. 

Line 30: tumour "in" the pediatric population. 

Line 31: The readers of this journal may or may not be familiar with modern medical history. The authors may remind here that Bailey & Cushing used surgery and it would be interesting to review how many patients reported in 1925 might have had a gross total resection.  

Lines 36-39: Has the discovery of molecular subgroups of MB been 20 years? Please confirm if the statement of "two decades" is true. 

Lines 137-141: May the authors be more specific here about their comment/suggestion on which type/style of chemotherapy should be used in WNT-MB? 

Lines 346-348: Intensive, frequent intrathecal/intraventricular/intraOmmaya chemotherapy is a key component of the MEMMAT protocol and should be stressed here. The feasibility of perfomming intensive IT/IVent chemotherapy in LMIC settings may also be discussed. 

Comments on the Quality of English Language

Moderate editing of English language required

Author Response

Reviewer 4

We thank Reviewer 4 for the positive comments on our work and the suggestions that will definitely contribute to improve the manuscript. Many of the questions from reviewer 4 relate to medulloblastoma management in LMIC. While we agree that this topic is very important, it is challenging – not to say impossible – to combine the 2 topics (HIC and LMIC) in the same manuscript. There would be contradictions for each statement (second surgery recommended for incomplete resection – however, second look surgery is unaffordable by families in most LMIC; access to radiotherapy is recommended within 4 weeks, however, waiting time in many LMIC is > 3 months…). We are fully aware of this and are planning to have a special manuscript on this topic.

This is a well-written article that provides a thorough review of the developmental history of chemotherapy in medulloblastoma with a global perspective. 

Major suggestions: 

  • Would this paper about cancer treatment be more suitable to a cancer journal rather than a general dianostic journal? The authors may discuss this issue with editors and the publisher

This manuscript is an invited paper, part of a special edition on paediatric brain tumors. The title was part of the invitation.

  • The authors may explain why they chose to focus on irradiation-sparing approaches. Was this due to limited access to radiation therapy in several settings? Would many families chose no radiation therapy because of unaffordable cost or the difficulties of survivorship (e.g. raising a surviving child/adolescent/adult with neurocognitive impairments) in LMICs? Backgrounds that facilitated this review may be discussed. 

Thank you for this important suggestion. We have added a background at the beginning of this section. As far as medulloblastoma in infants and young children are concerned, the literature in LMIC is very limited, probably because most of these children are not treated.  

  • Line 17: Define "high-dose" radiotherapy. 

Thank you. We have added a comment on the reduction of the dose of craniospinal irradiation in the section average risk. We hope that this answers the question.

  • Lines 19-20: Does "increasing complex condition" mean medulloblastoma? If so, the word "condition" may be changed into "disease". If the authors want to raise other issues, e.g. survivorship of MB, more information should be provided to explain the complexity of MB management. 

 We have changed the term as recommended.

  • More diagnostic aspects related to MB treatment should be discussed. Can the molecular subgroups be identified by pathologists in LMIC settings? Are there regional collabrative networks that may help to establish the molecular diagnosis (or a diagnosis that is compatible with the WHO 2021 scheme)? 

The issue of medullboastoma in LMIC is huge and would require a full paper. In this manuscript, we have included a small section on LMIC and tried to address several points raised by reviewer 4 including the molecular subgrouping.

  • Would the authors consider the recent discovery of sub-subgroups of MBs (e.g. alpha to delta of SHH MB) is clinically relevant? Please provide your point of view with citation(s). 

We have added a small paragraph on the subgrouping of medulloblastoma with these details.

  • Although it's a paper of systemic therapy in MB, the importance of neurosurgery and radiation therapy should still be briefly discussed here or in earlier paragraphs. The impact of gross total resection and near-total resection should be reviewed. Studies from LMICs would of particular value. 

We have added a section “principles of the management of pediatric medulloblastoma”

The authors may also discuss about - 

why young and old children with MB have been treated differently? why are there different age cutoffs in MB protocols and what are the commonly used cutoffs in LMICs? 

This issue is discussed in the section “principles of the management of pediatric medulloblastoma”

  • what are the patterns of radiation therapy for children with MB in LMICs? 

 This is an important question and it would be important to discuss this topic in the context of a specific paper on LMIC. However, we think that this question is beyond the scope of this paper on systemic therapies

  • Section #6 nicely reviewed the discovery of systemic therapy contributed by studies in LMICs. May the authors also address other issues in resource-limited settings, e.g., abandonment, treatment refusal, misdiagnosis, patient access to diagnosis and/or follow-ups, and toxic deaths? 

We agree this is a huge topic and would deserve a full paper. We did add these issues in the paragraph on LMIC.

Minor points: 

  • Which author(s) is/are from Affiliation (4)? 

Eric Bouffet

  • The affiliation of the corresponding author was not provided. 

Affiliation is provided

  • Line 30: tumour "in" the pediatric population. 

 This has been added

  • Line 31: The readers of this journal may or may not be familiar with modern medical history. The authors may remind here that Bailey & Cushing used surgery and it would be interesting to review how many patients reported in 1925 might have had a gross total resection.  

Thank you for raising this point. There were 4 radical resections reported in the original paper. This has been added in the text.

  • Lines 36-39: Has the discovery of molecular subgroups of MB been 20 years? Please confirm if the statement of "two decades" is true.

We agree with reviewer 4, that the seminal publications on subgrouping were in 2011-2012, and we have changed the sentence to “Last decade” 

  • Lines 137-141: May the authors be more specific here about their comment/suggestion on which type/style of chemotherapy should be used in WNT-MB? 

Thank you for this point. We have changed the sentence to “Maintenance chemotherapy with high cumulative doses of cyclophosphamide was a significant predictor of survival”

  • Lines 346-348: Intensive, frequent intrathecal/intraventricular/intraOmmaya chemotherapy is a key component of the MEMMAT protocol and should be stressed here. The feasibility of perfomming intensive IT/IVent chemotherapy in LMIC settings may also be discussed.

A comment on the potential of MEMMAT and the limitations as suggested by reviewer 4 has been added in the section LMIC 

Reviewer 5 Report

Comments and Suggestions for Authors

Very good review summarizing progress in the treatment regimens of different types of medulloblastoma over last half century. Current efforts to optimize the management of medulloblastoma are often an isolated attempts from different countries and any efforts to put together some sort of summary and develop common protocols are greatly appreciated.

I'd recommend to publish this paper with minor corrections: 

1) Line 4: "Syed Ahmer Hamid" is underlined and is a hyper-link, please remove. 

2) Line 4: Add affiliation for Eric Bouffet (4).

3) Line 52: change "...chemotherapy can find found..." to "...chemotherapy can be found..."

4) Line 336: "55% 3 year survival versus 38%" -- is there p-value available for these percentages?

5) Line 577: remove "7."

That it. Great job! 

Author Response

Reviewer 5

We are extremely grateful to Reviewer 5 for the positive comments and would like to thank him/her for the very kind remarks.

We have taken into account the recommendation for minor corrections

- Line 4: "Syed Ahmer Hamid" is underlined and is a hyper-link, please remove.

This has been removed 

- Line 4: Add affiliation for Eric Bouffet (4).

Affiliation added

- Line 52: change "...chemotherapy can find found..." to "...chemotherapy can be found..."

This has been changed.

  • Line 336: "55% 3 year survival versus 38%" -- is there p-value available for these percentages?

Thank you for the question, the p value (p = 0.007) has been added.  

  • Line 577: remove "7."

This has been removed.  

Round 2

Reviewer 1 Report

Comments and Suggestions for Authors

I thank authors to revise and to resubmit the article. Although the contents of article is certainly improved, it is not match for Diagnostics as I made a comment in the previous review. Therefore, I still stay in "reject" for this specific article as a reviewer of Diagnostics. I told the Editorial Office to transfer this article to an appropriate journal.

Author Response

We thank Reviewer 1 for his comment. While we understand that the manuscript does not seem to fit with the focus of the journal Diagnostics, this work is part of a series entitled “Medulloblastoma – Existing and Evolving Landscape”. We were invited to write a review on “Evolution of systemic therapy in medulloblastoma including irradiation-sparing approaches” and did not have any control over the topic assigned to us. We hope that Reviewer 1 will understand that the final decision is out of our hands in the context of this special issue.

Reviewer 3 Report

Comments and Suggestions for Authors

The manuscript improved greatly after the review carried out by the authors but still needs to improve the grammar. 

Comments on the Quality of English Language

The text still needs a small grammatical revision

Author Response

Thank you for your positive comments on our revised manuscript. We have taken into account the recommendation to improve the grammar and have done our best to address this issue.